# Effect of MyMAFI—A Newly Developed Mobile App for Field Investigation of Food Poisoning Outbreak on the Timeliness in Reporting: A Randomized Crossover Trial

**DOI:** 10.3390/ijerph16142453

**Published:** 2019-07-10

**Authors:** Fathul Hakim Hamzah, Suhaily Mohd Hairon, Najib Majdi Yaacob, Kamarul Imran Musa

**Affiliations:** 1Department of Community Medicine, School of Medical Sciences, Universiti Sains Malaysia, Kelantan 16150, Malaysia; 2Unit of Biostatistics and Research Methodology, School of Medical Sciences, Universiti Sains Malaysia, Kelantan 16150, Malaysia

**Keywords:** mobile application, field investigation, food poisoning outbreak, timeliness, reporting, MyMAFI

## Abstract

Prompt investigation of food poisoning outbreak are essential, as it usually involves a short incubation period. Utilizing the advancement in mobile technology, a mobile application named MyMAFI (My Mobile Apps for Field Investigation) was developed with the aim to be an alternative and better tool for current practices of field investigation of food poisoning outbreak. A randomized cross-over trial with two arms and two treatment periods was conducted to assess the effectiveness of the newly developed mobile application as compared to the standard paper-based format approach. Thirty-six public health inspectors from all districts in Kelantan participated in this study and they were randomized into two equal sized groups. Group A started the trial as control group using the paper-format investigation form via simulated outbreaks and group B used the mobile application. After a one-month ‘washout period’, the group was crossed over. The primary outcome measured was the time taken to complete the outbreak investigation. The treatment effects, the period effects and the period-by-treatment interaction were analyzed using Pkcross command in Stata software. There was a significant treatment effect with mean square 21840.5 and its corresponding F statistic 4.47 (*p*-value = 0.038), which indicated that the mobile application had significantly improve the reporting timeliness. The results also showed that there was a significant period effect (*p*-value = 0.025); however, the treatment by period interaction was not significant (*p*-value = 0.830). The newly developed mobile application—MyMAFI—can improve the timeliness in reporting for investigation of food poisoning outbreak.

## 1. Introduction

Food poisoning, which sometimes also known as foodborne illness, foodborne disease or foodborne infection, is a common communicable disease. In certain cases, it could be costly despite of the fact that it is preventable [1]. In 2010, it was estimated that about 600 million people globally were affected by foodborne diseases, with 420,000 deaths reported [2]. About 40% of the foodborne diseases burden affecting young children under 5 years old of age with 125 000 deaths recorded annually [2]. Higher incidence was noted among the population who live in low-income countries [3]. According to Havelaar et al. (2015), the global burden of foodborne diseases is almost equivalent to other major communicable diseases such as tuberculosis, malaria and HIV [4]. In Malaysia, the incidence of foodborne diseases ranged from 0.14 to 1.56 cases per 100,000 populations [5]. The actual number of people affected and burden of the disease could be much greater as there are still many food poisoning cases or outbreak that were not reported, and hence not investigated [5].

Field investigation is one of the crucial aspects in epidemiological investigation which the primary aim is to identify the source of a problem based on the suggestive evidences or information gathered throughout the process [6]. The current practice of food poisoning outbreak investigation in Malaysia involves multi-stages of processes. After receiving a notification, the public health inspector (PHI) will inform the case to Medical Officer of Health and verification of the incident will be done. Once verified, an initial investigation can commence. Based on the findings from the initial investigation, the PHI will prepare to submit the initial report to State Health Department (JKN). In order to ensure prompt outbreak investigation being initiated, Malaysia Ministry of Health has set the timeline indicator for reporting. The initial report must be submitted to JKN within 20 hours and to Ministry of Health (MOH) within 24 hours from the time of outbreak is declared. In the initial report, the outbreak control team must identify the suspected source of food poisoning outbreak, which is the contaminated food most likely to have contributed to the occurrence of the outbreak. Preventive action and control measures will be carried out at the field regardless whether the detailed investigation in identifying the source of the problem has been completed or not. Lastly, a final report must be submitted to the JKN within one month from the date the outbreak was declared [7].

The current procedure of field investigation is using the conventional approach which is written paper format. Basically, there are two sets of investigation forms; the first one is the investigation form to collect the information of the outbreak episode as a whole, such as date and time it was notified, its location and the number of people affected. The second investigation form is the form used to collect information about food intake history from every individual case that was involved in the outbreak. The current practices require two phases of manual data entry and calculation, where the first phase is in the field during the investigation and the second phase is when the PHI returns to district health office to key in the information into a computer. In order to determine the source of infection, they need to prepare a line-listing of cases and controls with types of meals that were consumed. From the line-listing, they will analyze the data to get the odds ratio (OR). A high OR value is one of the important parameters in determining the source of food contamination. This whole process increases the likelihood of errors in data entry and it is also time consuming. Consequently, it may lead to a delay in reporting and hence a delay in implementing efficient prevention and control activities, such as withholding the supply of raw foods, which are the potential source of contamination.

Most cases of food poisoning involve a short incubation period. Thus, timeliness of reporting is a crucial factor that influences the implementation of effective public health interventions [8]. The key strength of an outbreak investigation is determined by how rapid the public health acts and initiates coordination across multiple organizations [9]. An efficient field investigation approach would lead to a more rapid and facilitated implementation of preventive and control measures that would break the transmission of the outbreak in a shorter time. Utilization of technological advancement for outbreak investigation is one of the approaches that could improve the time to response to outbreak [10]. Thus, this study aims to assess the effectiveness of a newly developed mobile application (mobile app) as an alternative tool for field investigation of food poisoning outbreak in reducing the time taken for reporting.

## 2. Materials and Methods

### 2.1. Study Settings and Participants

This study was a randomized cross-over trial with two arms and two treatment periods, which was conducted state-wide between September 2018 and November 2018 in Kelantan. Kelantan is a state situated in north east region of peninsular Malaysia. With an anticipated 10% drop-out, the calculated sample size to study the differences in the time taken to produce report was 37 per group. A total of 36 PHI from all districts that fulfill the inclusion and exclusion criteria participated in this study. The inclusion and exclusion criteria were as listed below:

Inclusion Criteria:Public health inspectors from food and water-borne disease unit or involve with food poisoning outbreak investigation.Attended the two-day introductory course for MyMAFI (My Mobile Apps for Field Investigation).Own a smartphone that used the Android platform.

Exclusion Criteria:Public health inspectors who newly joined the Food and Waterborne Diseases Unit and never experience food poisoning outbreak investigation.

They were randomly assigned to either group A or group B. Randomization of the participants was performed using the ‘rand()’ function in Microsoft Excel 2013 (Microsoft, Redmond, WA, USA) for Windows. From the list of participants in Excel file, the ‘rand()’ function assigned a random number between 0 to 1 for each of the participants and then the numbers were sorted from smallest to largest. The first half in the list were assigned to group A and the second half were assigned to group B. The randomization process was implemented by one of the research team members.

The study tools include a newly developed mobile app for field investigation of food poisoning outbreak known as MyMAFI (My Mobile Apps for Field Investigation), which was used as an intervention. It is a mobile app that used Android platform and the size of installation file is 3.3 megabytes. MyMAFI which was recently developed by our research team members is basically the conversion of the standard investigation form into a mobile app but with a few add-on features such as auto-calculation for certain descriptive statistics, auto-generation of epidemic curve and auto-generation of line-listing of cases and controls. Other tools were paper format investigation forms, which were used by control group, where there were two sets of forms. The first one was the Investigation Form/Report for Food Poisoning Outbreak for documentation of the general information about the outbreak such as details on the notification, types and address of the premises implicated and a descriptive statistics on cases involved with the outbreak [7]. The second form is the Investigation Form for Food Poisoning—Individual Case for documentation of the information about every individual case, such as the sociodemographic information, contact details, signs and symptoms and more importantly the information on foods history.

The first phase of outbreak simulations was conducted two weeks after the introductory course on MyMAFI. The introductory course was held for all PHI from Food and Water-Borne Disease Unit from all district health offices in Kelantan. The course was conducted over two days and involved learning theoretical aspect of MyMAFI function and how it works as well as a practical or hands-on session. In the first phase of the intervention, the PHI in group A started with the conventional paper-based forms for investigation of simulated food poisoning outbreak; meanwhile, the PHI in group B used the intervention which is MyMAFI as their tool. Both groups used Set I outbreak simulations which contains 18 different scenarios of outbreaks. This means that every single PHI in each group conducted one outbreak investigation which differed to the rest of their group members. After a one-month washout period, the second phase was conducted. The groups were crossed over for method of investigation where group A used MyMAFI and group B used the standard paper format and this time they used Set II outbreak simulations which consist of another 18 different outbreak scenarios. Each scenario was created and adapted from previous food poisoning outbreaks in Kelantan. Two research team members had been assigned with the task to create and prepare these outbreak simulations. Figure 1 shows the flow chart of the study.

For this outbreak simulation, it is assumed that the outbreaks have been notified and verified. All participants had been reminded to bring their own laptop beforehand, as those who are assigned to the control group had to use it to input the information from the investigation form that they filled in. They were considered as having completed the outbreak investigation once all the forms had been filled up. They were also required to submit a complete line-listing of cases and control. They were instructed to notify the researcher instantly once they completed their investigation, after which the researcher checked to confirm whether the information was complete or not; only then would the time of completion be documented in minutes.

### 2.2. Data Entry and Analysis

For data entry and analysis, Group A was labeled as sequence 1 and group B was labeled as sequence 2. Important baseline characteristics among participants between groups were compared using independent T-test for numerical variables. As for categorical outcomes, Fisher Exact Test was used, because two cells (50.0%) had expected frequencies less than 5. The baseline characteristics that were being compared were age of participants, gender and their service grade, which was classified either junior or senior grades. Set I and Set II of the outbreak simulations were also compared in terms of number of cases affected in each outbreak and number of meals/foods that associated with it as these two variables are the potential strong confounders that can affect the primary outcome of this study, which is the timeliness in reporting. Independent T-test was used for comparison between Set 1 and Set II of outbreak simulations.

Mean timeliness in reporting during treatment period 1 and treatment period 2 were calculated for each group and were compared according to the treatment period using independent T-test. Statistical analysis for cross-over trial was performed by using the Pkcross package analysis in Stata software version 14 (StataCorp, College Station, TX, USA) to estimate the overall mean, the treatment effects, the period effects and the period-by-treatment interaction while assuming that there is no carry-over or sequence effect exist. A p-value of less than 0.05 is taken as the cut-off point to consider a statistically significant result.

### 2.3. Ethical Considerations

This study has been approved by the Human Research Ethics Committee USM, Malaysia (USM/JEPeM/ 17120673) and National Medical Research Registry, Malaysia (NMRR-17-2852-38654).

## 3. Results

### 3.1. Sociodemographic Characteristics of Participants

The mean (standard deviation) age of participants in group A was 40.1 (8.58) years whereas the mean (SD) age for Group B was 36.9 (77.7) years. The majority of participants in both groups were males (72.2% in group A versus 88.9% in Group B). More than two third of participants in both groups were among junior grades (77.8% in Group A and 88.9% in Group B). Comparative analysis between groups showed there were no significant differences in terms of age, gender and grade seniority (Table 1).

### 3.2. Outbreak Simulations

There were two potential confounding variables that could affect the outcomes of this study: number of cases and number of meals involved. These were checked and Independent T-test results showed that there were no significant differences between Set I and Set II outbreak simulations (Table 2).

### 3.3. Effectiveness of MyMAFI on the Timeliness in Reporting

Analysis on the reporting speed between the two groups according to the treatment period was done. During treatment period 1, the mean differences between the two groups was 31.2 minutes, where group sequence 2 (used MyMAFI in treatment period 1) showed a shorter timeline as compared to group sequence 1 (used control method in treatment period 1), although it was not statistically significant. The same trends noted during treatment period 2 when the groups were crossed over; group sequence 1 which used MyMAFI during this period showed improved reporting timeliness with mean differences of 38.4 minutes (Table 3).

Further analysis using Pkcross package in Stata looking at the effect of MyMAFI, the period effect and the period-by-treatment interaction showed that MyMAFI mobile app had significant effect on the reduction of reporting timeliness. It was also found that there was significant period effect, but the period-by-treatment interaction turned out to be non-significant (Table 4).

## 4. Discussion

The researchers’ decision to conduct a cross-over study instead of a parallel design was made based on a few factors. Firstly, the occurrence of food poisoning outbreak is unpredictable, and the incidence is also low. Therefore, we decided to test MyMAFI as intervention via outbreak simulations, but another issue raised here as the eligible PHI that fulfill the inclusion and exclusion criteria for this study were only 36 persons. With two-arm design (intervention versus control), the samples would be too small if we proceeded with parallel design. One of the known advantages of cross-over trials is that it requires a smaller sample size as compared to the conventional parallel-group design [11,12].

A second factor that made us to choose the cross-over design was because there are multiple known and unknown confounding variables that can affect the timeliness in reporting. Confounders that can be objectively measured, such as age of participants or number of cases involved in each outbreak, might not be a problem, as it can be handled statistically; however, for more subjective confounders, such as the efficiency of a PHI in conducting, the outbreak investigation would be a troublesome process that can jeopardize the validity of this study results. By conducting cross-over trials, the between-subject variability can be minimized or eliminated [12,13], and hence, the effect of the intervention on the outcomes can be exclusively ascertained [14].

Nevertheless, there are certain issues that have to be taken into consideration when conducting a cross-over trial and one the main concerns is about the carry-over effect. Senn (2002) in his book defined the carry-over effect as “the persistence (whether physically or in terms of effect) of a treatment applied in one period in a subsequent period of treatment” [15]. According to Senn, the carry-over effect is more likely to exist when there is no wash-out period. The wash-out period for pharmacotherapy interventions varies, as it depends on the type of drugs and its half-life. Meanwhile, for lifestyle intervention, for example, a cross-over study which investigated the effect of high intensity interval training (HIIT) on the clinical outcomes of heart transplant recipients compared to the continued moderate exercise reported that the wash-out period required was up to 5 months without exercise in order to lose the effect of the intervention [16]. In our study, we can say that there is no or minimal carry-over effect, as the intervention that we tested was neither a pharmacotherapy nor a lifestyle intervention that may have had residual effects.

Another issue that is associate with the cross-over trial is the possibility of treatment by period interaction, which is related to the carry-over effect [17]. This situation is more likely to occur in a cross-over trial that involves long treatment periods with unstable medical conditions [11]. The possibility of treatment by period interaction in our study is minimal, as our intervention was a one-off basis for each treatment period and the condition that is being studied is also stable, as it involved an outbreak simulation that have been prepared earlier on.

### 4.1. Characteristics of the Participants

Randomization of PHI into two groups led to groups that were comparable in terms of baseline characteristics which include age, gender and grade of service. Seniority in services among PHI can be assumed to be a proxy variable for level of training, which cannot be objectively measured. Significant difference in this variable may alter the outcomes of this study as it is a potential confounder; lack of training has been proven to be associated with non-timeliness [18,19]. There are other potential confounders related to participants’ characteristics that cannot be measured, such as their familiarity in using mobile apps, as using its features on smartphones require certain level of manual dexterity and those with small-screen smartphones have to possess good visual acuity to visualize its interfaces [20]. Realizing this issue, in our study, participants were instructed to install and use MyMAFI on their own smartphones, as they are already familiar with them.

### 4.2. Timeliness in Reporting

In both treatment periods, the results revealed that the group that used MyMAFI took a shorter time to complete the investigation as compared to the group that used the conventional paper-format; however, the mean differences in the time taken were not statistically significant. However, when the overall results were analyzed using Pkcross for cross-over experiments in Stata, there is a significant treatment effect, with a mean square of 21,840.5 and its corresponding F statistic 4.47 (*p*-value = 0.038). The reduction in timeliness of reporting when using the MyMAFI was around 18%. A study conducted in South Africa showed that the timeliness in reporting was shortened by 81% when the paper-based format for reporting was converted to the use of mobile phones [21]. A pilot study on the use of mobile phones for syndromic surveillance conducted in Papua New Guinea reported bigger changes in timeliness of reporting, as the new system reduced the timeline from 84 days to 2.4 days, which is equivalent to a 97% reduction in reporting timeliness [22].

There are a few reasons that could explain why the reduction in reporting timeline in our study is not as marked as the findings in the studies by Quan et al. and Rosewell et al. The study by Quan et al. was conducted in a rural setting area and in a real-life situation. It is possible that reduction in travelling time plays a major role when the reporting system was changed from paper-based format to mobile phones. Moreover, it is possible that travelling time also plays a major role in the study by Rosewell et al., which was also conducted in a real-life situation. Furthermore, 87% of Papua New Guinea population lives in rural places; thus, access to healthcare services is one of their major issues [22]. On the other hand, our study was conducted under a ‘controlled situation’, which is via centralized outbreak simulations; hence, travelling time is not a determinant for the outcomes in our study. In addition, the nature of infectious disease that are being evaluated also plays a role in influencing the timeliness. Quan et al. studied on malaria whereas we focused on food poisoning. In food poisoning, detailed information on food history must be obtained, which is quite time-consuming. Moreover, it is difficult to make a direct comparison in timeliness of reporting between countries, as each country has their own regulations for surveillance system, the complexity of reporting procedures are also different and the required timeframe for reporting also vary [23].

Nevertheless, there is a convincing evidence that electronic reporting is more effective as compared to the conventional paper-based format; this finding is supported by a systematic literature review study conducted by Swaan et al., which shows that electronic reporting can shortened the reporting timeliness from days up to months faster, and findings from another systematic literature review conducted by Steele et al. [24,25]. Another study, which introduced the use of a mobile app known as AREA (A Mobile Application for Rapid Epidemiology Assessment) that was designed for epidemiology assessment, revealed that the mobile app managed to facilitate rapid course of action and coordination of resources [26]. AREA can be used in situations such as public health crises or humanitarian crisis management, but it is not tailored for specific management of infectious disease outbreak.

It is noted that in our study results, there is a significant period effect (*p*-value = 0.025). It is possible that this is due to a different set of outbreak simulations that were used for the second treatment period in our study. The reason why we used a different set of outbreak simulations in the second period was because we wanted to avoid a learning effect bias. If the same outbreak simulations are being used, it is possible that the participants still remember some of the information that they had documented during the previous treatment period, and hence, the process of entering the information during the second treatment period would be faster. Though we used a different set of outbreak simulations, we did compare between the two sets and analysis on its potential confounding variables which are number of cases and number of meals in each set showed no significant differences (Fisher Exact Test: *p*-value = 0.154 and 0.903, respectively). The treatment by period interaction, which was not significant (*p*-value = 0.830), indicated that the intervention did not work differently despite of the differences in the outbreak simulations between the study period.

### 4.3. Strengths, Limitation and Recommendation

The utilization of mobile apps for the surveillance of infectious diseases in Malaysia is a new area that should be explored. To the best of our knowledge, this is the first study conducted to assess the effectiveness of a newly developed mobile app for field investigation of outbreak. Moreover, the assessment of the effectiveness of the mobile app was done via a randomized cross-over trials, a powerful study design that can minimize the between-individuals variability, and hence, the true impact of the mobile app on the study outcome, which is the timeliness in reporting, can be exclusively evaluated. MyMAFI mobile app also has been registered for intellectual property under Intellectual Property Corporation of Malaysia (Intellectual Property Number: LY2019001337).This study also had limitations, which include the assessment on the effectiveness of the mobile app as compared to the conventional paper-based format that was conducted via outbreak simulations and not during the real-life outbreaks. The effect on the timeliness between the two methods would be different if the trial is conducted under real food poisoning outbreaks, as other potential confounding factors have to be taken into consideration such as accessibility to the outbreak locations and more importantly the recall bias that might present when investigating the foods intake histories from cases.

There was also the possibility of non-response bias, as at recruitment stage, there were PHIs who declined to participate in this study. Furthermore, the characteristics between participants and non-participants could be systematically different, which would lead to non-response bias [27]. For example, it is possible that those PHIs who declined to participate are the one who are less tech-savvy, which might mean better outcomes by using a mobile app only hold up among the tech-savvy individuals.

For a better assessment on the effectiveness of MyMAFI on the reduction in timeliness of reporting, it should be tested and applied for field investigation of real food poisoning outbreaks and not just through simulations. Assessment on its effectiveness should not just based on the quantification of reduction of the reporting timeline alone, but should also look at the numbers of cases that can prevented. This is highly important and reflects a more efficient investigation and response.

In terms of the product, namely the mobile app itself, MyMAFI can only cater for the Android platform and it is only available in Malay. Thus, any subjects who use the iPhone Operating System (iOS) or Windows-based smartphones cannot use the mobile app and were not included in this study. There might be differences in characteristics of PHI among those who use android platforms with those who use iOS or Windows platforms, which could affect the outcomes of this study. As a recommendation, the use of MyMAFI should be expanded to cater all major operating systems and platforms and it should be made available in a dual language medium (Malay and English).

## 5. Conclusions

This trial revealed that MyMAFI is effective as an alternative tool for field investigation of food poisoning outbreak as the time taken for reporting by using this mobile app was less than the conventional method. Nevertheless, its effectiveness during an actual outbreak is yet to be tested. In addition to that, although the reduction in reporting time was only around 30 minutes, there are other intangible advantages of using MyMAFI. It is a step forward towards a paperless approach in public health surveillance and it could also reduce the workload of PHIs, as the current practices require them to perform redundant tasks in manual data entry.

## Figures and Tables

**Figure 1 ijerph-16-02453-f001:**
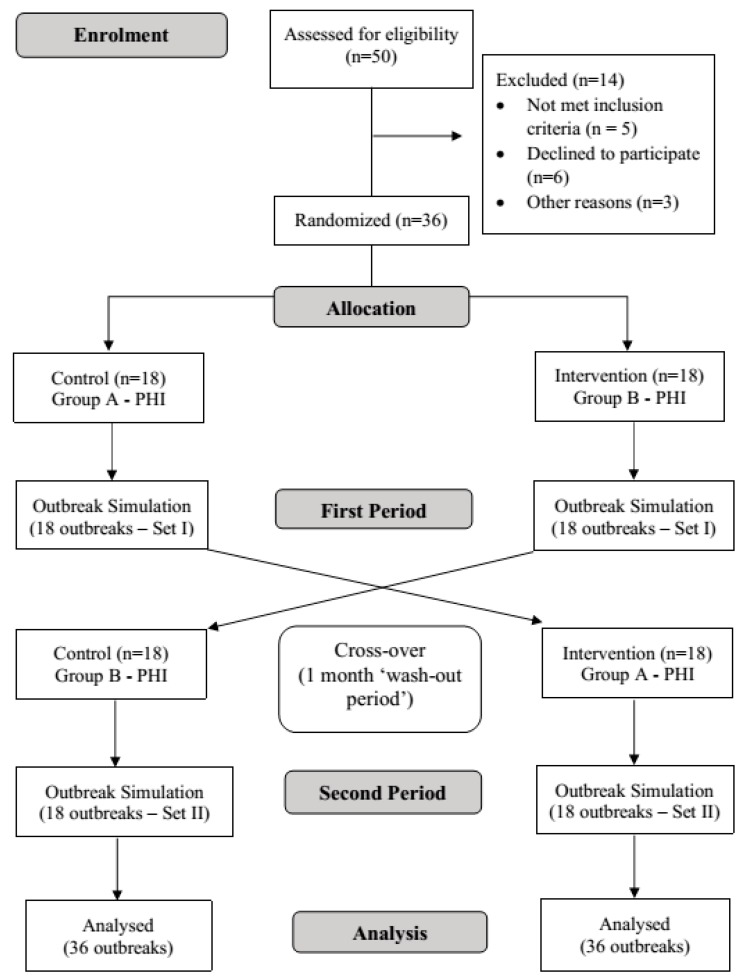
Flow chart of this study.

**Table 1 ijerph-16-02453-t001:** Comparison of baseline participants’ characteristics between group. MyMAFI: My Mobile Apps for Field Investigation.

Variables	Group A (Sequence Control—MyMAFI) (*n* = 18)		Group B (Sequence MyMAFI—Control) (*n* = 18)		*p*-Value
	*n* (%)	Mean (SD)	*n* (%)	Mean (SD)	
Age (years)		40.1 (8.58)		36.9 (7.77)	0.254 ^a^
Gender					0.402 ^b^
Female	5 (27.8)		2(11.1)		
Male	13 (72.2)		16 (88.9)		
Grade					0.658 ^b^
Junior	14 (77.8)		16 (88.9)		
Senior	4 (22.2)		2(11.1)		

^a^ Independent T-Test; ^b^ Fisher Exact Test.

**Table 2 ijerph-16-02453-t002:** Comparison of number of cases and number of meals between Set I and Set II of outbreak simulations.

Variables	Outbreak Simulation Mean (SD)		Mean Difference (95% CI)	t−Statistic (d.f.)	*p*−Value
	Set I (*n* = 18)	Set II (*n* = 18)			
Number of Cases	9.8 (4.41)	11.9 (4.27)	−2.11 (−5.05, 0.83)	−1.46 (34)	0.154
Number of Meals	12.8 (4.99)	13.1 (5.82)	−0.22 (−3.89, 3.45)	−0.12 (34)	0.903

d.f.: degree of freedom.

**Table 3 ijerph-16-02453-t003:** Comparison of timeliness in reporting between groups according to treatment period.

Treatment Period	Group Sequence Mean (SD)		Mean Difference (95% CI)	t−Statistic (d.f.)	*p*−Value
	1 (*n* = 18)	2 (*n* = 18)			
1	168.9 (77.67)	137.6 (59.46)	31.2 (−15.58, 78.13)	1.36 (34)	0.184
2	171.8 (64.58)	210.2 (76.11)	38.4 (−9.42, 86.20)	1.63 (34)	0.112

**Table 4 ijerph-16-02453-t004:** Effects of intervention, its period effect and period-by-treatment interaction on the timeliness in reporting.

Variables	MS	F-Statistics	*p*-Value
Treatment effect	21,840.5	4.47	0.038
Period effect	25,613.4	5.25	0.025
Treatment*Period	227.6	0.05	0.830

**MS: Mean Squares.**

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
