# Peer review of "Effect of MyMAFI—A Newly Developed Mobile App for Field Investigation of Food Poisoning Outbreak on the Timeliness in Reporting: A Randomized Crossover Trial"

_ijerph, 2019, doi:10.3390/ijerph16142453_

Round 1
Reviewer 1 Report
In the assessment of the paper submitted for the review, I specifically focused on the discussed issues, applied research procedure, substantive content of the paper and its structure.
The subject area discussed in the paper is important and topical. It is also consistent with the profile of the Journal.
The considerations conducted in the paper are focused on such categories as: mobile application, MyMAFI, field investigation, food poisoning outbreak, timeliness, reporting.
The value of the paper results from appropriate combination of literature studies with the results of an empirical research.
The structure of the paper is clear.
However, deliberations conducted in the paper need to be expanded. Therefore, it is specifically recommended to:
- take into consideration more publications in the sphere of discussed subject matter,
- develop the conclusions,
- provide in-depth the description of research limitations and product (MyMAFI) limitations.
Reviewer 2 Report
Reviewer comments and suggestions to the authors
Manuscript ID: ijerph-543515
The present manuscript describes the use of a mobile technology, named as MyMAFI as a tool for current practices for field investigation on food poisoning outbreak. Throughout this study, the authors compared the effectiveness of this newly technology with the standard paper‐based format approach.
Next are some comments and suggestions that may be useful for improving this manuscript.
- Please check the punctuation and spaces in the text.
- Check the text in terms of language, keeping a constant verbal tense.
-The novelty and the advantages of this study must be strongly supported. The authors must emphasize the news of it.
- Are the authors aware of other technological applications on this field?
-The validation of final conclusions must be strongly emphasized with a broader discussion around this issue.
FINAL COMMENTS AND CONSIDERATIONS: It deserves to be published after the suggestions and corrections listed above are amended in order to improve the manuscript.
Reviewer 3 Report
The work is based on a solid experimental design and on a well-conducted statistical elaboration. The text describes the work completely. For completeness, I would advise to indicate in table 1 the meaning of the reported values, as done in tables 2 and 3: “Mean (SD)”.
